# High-fidelity photonic quantum logic gate based on near-optimal Rydberg single-photon source

Shuai Shi[1,2], Biao Xu[1,2], Kuan Zhang[1,2], Gen-Sheng Ye [1,2], De-Sheng Xiang[1], Yubao Liu[1], Jingzhi Wang[1], Daiqin Su [1] ✉ & Lin Li [1] ✉

Compared to other types of qubits, photon is one of a kind due to its unparalleled advantages in long-distance quantum information exchange. Therefore, photon is a natural candidate for building a large-scale, modular optical quantum computer operating at room temperature. However, low-fidelity two-photon quantum logic gates and their probabilistic nature result in a large resource overhead for fault tolerant quantum computation. While the probabilistic problem can, in principle, be solved by employing multiplexing and error correction, the fidelity of linear-optical quantum logic gate is limited by the imperfections of single photons. Here, we report the demonstration of a linear-optical quantum logic gate with truth table fidelity of 99.84(3)% and entangling gate fidelity of 99.69(4)% post-selected upon the detection of photons. The achieved high gate fidelities are made possible by our near-optimal Rydberg single-photon source. Our work paves the way for scalable photonic quantum applications based on near-optimal single-photon qubits and photon-photon gates.

Entangling operation is one of the fundamental building blocks for universal quantum computation[1,2]. A probabilistic, heralded two-photon quantum logic gate is sufficient to achieve scalable linear-optical quantum computing by using the Knill–Laflamme–Milburn (KLM) scheme[3], albeit with a substantial resource overhead. The cluster-state model[4–6], which is based on the local measurement and feeds forward on a large entangled cluster state, can significantly reduce the resource overhead[7–9]. Given that implementing a deterministic two-photon entangling operation is still challenging in linear optics, a large entangled cluster state can be generated by fusing a collection of small clusters[7], e.g., three-photon clusters, in a ballistic way[8,10–12]. The resource consumption of the cluster-state computation is then dominated by the preparation of those small clusters[13], which depends on the efficiency and quality of single-photon sources, and the ability to generate entanglement with quantum logic gate, i.e., the photon–photon entangling gate fidelity.

After the initial proposal of KLM[3], a destructive optical controlled-NOT (CNOT) gate (without the entangling gate operation) was experimentally demonstrated in 2002, with a truth table fidelity of 83%[14]. Qiang et al. employed a different linear-optical scheme[15] with four pairs of entangled photons as input and achieved a gate with 98.85% truth table fidelity and 1/64 intrinsic success probability. The first entangling operation using a linear-optical gate was realized in 2003, with an intrinsic success probability of 1/9 and an entangling gate fidelity of 87%[16]. Optical-related errors, such as photon mode mismatch and polarization imperfection, are reduced over the years, and the entangling gate fidelity has been gradually improved to ~94%[17–19]. Now that many technical sources of error have been addressed, the major obstacle for further suppressing the entangling gate infidelity lies in the quality of single-photon sources[20]. For example, to achieve a linear-optical quantum logic gate with infidelity below 1%, the minimum requirements on a single-photon source are

[1]MOE Key Laboratory of Fundamental Physical Quantities Measurement, Hubei Key Laboratory of Gravitation and Quantum Physics, PGMF, Institute for Quantum Science and Engineering, School of Physics, Huazhong University of Science and Technology, Wuhan 430074, China. [2]These authors contributed equally: Shuai Shi, Biao Xu, Kuan Zhang, Gen-Sheng Ye. ✉e-mail: sudaiqin@hust.edu.cn; li_lin@hust.edu.cn

$g^{(2)}(0) < 7 \times 10^{-3}$ and the indistinguishability higher than 99%. To date, these demanding requirements have not been simultaneously achieved by the state-of-the-art single-photon sources.

Recently, significant progress has been made in single-photon sources based on cold Rydberg atoms[21–24]. The strong interactions between Rydberg atoms lead to excitation blockade[25] and hence the efficient preparation of single atomic excitation, which can be converted into high-quality single photon on demand[26] through matter-light quantum state transfer.

Here, we demonstrate a photon–photon quantum logic gate by performing the KLM CNOT gate protocol with single photons generated from Rydberg atoms. Our Rydberg single-photon source features near-optimal purity and indistinguishability and results in a high truth table fidelity of 99.84(3)% and entangling gate fidelity of 99.69(4)%.

## Results

### Experimental protocol

As illustrated in Fig. 1a, an ensemble of cold $^{87}$Rb atoms is prepared in an optical dipole trap and can be coupled from the ground state $|g\rangle$ to a high-lying Rydberg state $|r\rangle$ using a two-photon transition with a 780-nm laser field $\Omega^e_{780}$ and a 479-nm laser field $\Omega^e_{479}$ via an intermediate state $|e\rangle$. The waists of the Rydberg excitation and the dipole trap beams are chosen such that the entire excitation region is within the Rydberg blockade radius. As a result, multiple Rydberg excitations are suppressed and the entire atomic ensemble involving $N$ atoms can be promoted from the ground state $|G\rangle = \prod_{i=1}^{N} |g_i\rangle$ to the single collective excitation state $|R\rangle = \sum_{i=1}^{N} |g_1\rangle \ldots |r_i\rangle \ldots |g_N\rangle/\sqrt{N}$ by a $\pi$ pulse. To generate single photons on demand, a readout field $\Omega^r_{479}$ resonant with the $|e\rangle \leftrightarrow |r\rangle$ transition is applied. Assisted by the enhanced atom-light cooperativity, the readout field efficiently converts the Rydberg excitation into a single photon with a well-defined spatial mode.

To implement the CNOT gate[27–30], two single photons are consecutively generated from the Rydberg atoms and used as the control and target photonic qubits in a free-space photon–photon interferometer. As shown in Fig. 1b, the non-classical correlations between the control and target qubits can be established via the two-photon quantum interference at a partial polarization beam splitter (PPBS), which has 1/3 reflectivity for horizontally polarized photons and is totally reflective for vertically polarized photons. When the control qubit is in the vertically polarized state, encoded as $|0\rangle_C$, two photons do not interfere so the target qubit remains unchanged. In contrast, with the control qubit in the horizontally polarized state, encoded as $|1\rangle_C$, the horizontally polarized component of the target qubit acquires a $\pi$-phase shift as a result of the unbalanced two-photon quantum interference, while the vertically polarized component is unaffected. By encoding the diagonally and anti-diagonally polarized states as $|0\rangle_T$ and $|1\rangle_T$, the target qubit is flipped when the control qubit is in $|1\rangle_C$. Therefore, the implemented input-output relation is

$$|00\rangle \rightarrow |00\rangle, \qquad |01\rangle \rightarrow |01\rangle,$$
$$|10\rangle \rightarrow |11\rangle, \qquad |11\rangle \rightarrow |10\rangle,$$

which defines a CNOT gate operation.

### High-quality Rydberg single-photon source

The fidelity of the photon–photon quantum logic gate critically relies on the purity and indistinguishability of the Rydberg single-photon source. The purity of our Rydberg single-photon source is characterized by a Hanbury Brown–Twiss experiment, in which the photons are coupled into a 50:50 fiber beam splitter followed by two single-photon counting modules (SPCMs). Figure 2a shows the measured second-order intensity correlation function $g^{(2)}(\tau)$ as a function of delay $\tau$. As a

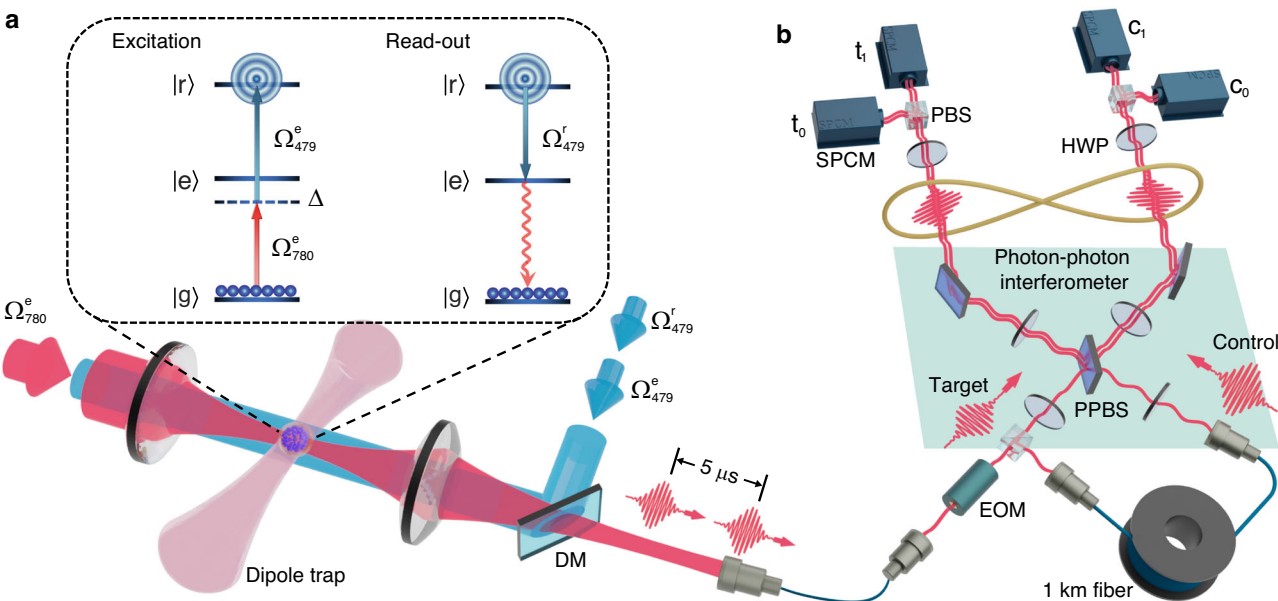

**Fig. 1 | Illustration of the experimental protocol. a** An ensemble of cold $^{87}$Rb atoms with an optical depth of ~5 is confined in a 1012-nm dipole trap. The counter-propagating 780 and 479 nm excitation beams are combined on a dichroic mirror (DM) and tightly focused onto the ensemble, with waists of 6 and 30 μm, respectively. Atoms are initialized in the ground state $|g\rangle$ and excited to a high-lying Rydberg state $|r\rangle$ with a single-photon detuning of $\Delta/2\pi = -200$ MHz. After the excitation, a 479-nm readout light resonant with $|r\rangle \leftrightarrow |e\rangle$ transition converts the single collective excitation state $|R\rangle$ into a single photon that is coupled into a single-mode fiber. The insert shows the atomic levels involved in the excitation and readout processes: ground state

$|g\rangle = |5S_{1/2}, F = 2, m_F = 2\rangle$, intermediate state $|e\rangle = |5P_{3/2}, F = 3, m_F = 3\rangle$, and Rydberg state $|r\rangle = |90S_{1/2} J = 1/2, m_J = 1/2\rangle$. **b** Two single photons are sequentially generated with an interval of 5 μs, and their temporal wave packets are well overlapped at the interferometer using a polarization switching electro-optic modulator (EOM) and a 1-km delay fiber. Before the two-photon interference at the first PPBS, two half-wave plates (HWPs) are employed to prepare the input state of the control and target qubits. Two more PPBSs and HWPs are used after the interference to complete the CNOT gate operation. The output state is measured by a polarization-sensitive detection setup consisting of HWPs, polarization beam splitters (PBSs), and SPCMs ($c_{0,1}$ and $t_{0,1}$).

result of the Rydberg excitation blockade, strong suppression of two-photon events at zero delay is observed. In order to achieve low $g^{(2)}(0)$, efforts are spent on suppressing the background detection events to SPCM dark-count level (see Supplementary Note 1). The measured value for the second-order intensity correlation function at zero delay is $g^{(2)}(0) = 7.5(6) \times 10^{-4}$, which indicates excellent single-photon purity.

The scalability of interference-based quantum photonic protocols strongly depends on the indistinguishability of single photons, since distinguishable photons severely deteriorate the operation fidelity. To investigate the indistinguishability, two-photon interference visibility is measured by a Hong–Ou–Mandel (HOM) experiment. Figure 2b displays the measured two-photon coincidence rate as a function of the temporal mismatch $\Delta t$ between two photons. The two photons used in the HOM experiment are sequentially generated from the Rydberg atoms. The first photon is delayed by 5 μs using an EOM and a 1-km fiber, while the second photon is generated 5 μs + $\Delta t$ after the first one (see Supplementary Note 2). Due to the quantum interference between two single photons, we observe a non-classical suppression of coincidences at zero temporal mismatch with a high visibility of $V = 99.43(9)\%$. Besides photon indistinguishability, $V$ is also affected by background detection events and single-photon impurity characterized by non-zero $g^{(2)}(0)$. By analyzing these contributions, we extract indistinguishability of 99.55(9)%.

We find that the reduction of indistinguishability from unity is mostly caused by the imperfect interference from photon components in the rising and falling edges of the single-photon temporal profile (see Supplementary Note 2). To further improve the indistinguishability, the single-photon detection window is reduced from 200 to 80 ns, which contains 62% of the photons around the center. The smaller detection window leads to a better indistinguishability of 99.94(8)% and is used for the CNOT gate experiment. The 80-ns detection window also results in a better signal-to-background ratio and a lower $g^{(2)}(0)$ of $4.5(8) \times 10^{-4}$.

## High-fidelity quantum logic gate

Having single photons with near-optimal purity and indistinguishability at hand, we proceed to the implementation and characterization of the photonic CNOT gate by inputting different control-target qubits combinations to the gate and measuring the corresponding output states. The truth tables of the CNOT gate are shown in Fig. 3a, b, where the post-selected probabilities for different input-output combinations are displayed on a logarithmic scale. The measured truth table fidelities of the CNOT gate in the computational ZZ basis and the complementary XX basis are $F_{\mathrm{CNOT}}^{\mathrm{ZZ}} = 99.84(3)\%$ and $F_{\mathrm{CNOT}}^{\mathrm{XX}} = 99.81(3)\%$, which is defined as the probability of detecting the desired output states averaged over input states.

The most remarkable feature of a two-photon quantum logic gate is its ability to establish entanglement between two initially uncorrelated photons. With a product state of $(|0\rangle - |1\rangle)|1\rangle/\sqrt{2}$ as input, the CNOT gate ideally generates a maximally entangled Bell state $|\Psi^-\rangle = (|01\rangle - |10\rangle)/\sqrt{2}$. To characterize the entangling gate fidelity, quantum state tomography is performed on the output state and the reconstructed density matrix $\rho$ is shown in Fig. 3c, d. The measured state fidelity, i.e., the entangling gate fidelity, is $F_{\Psi^-} = \langle \Psi^- | \rho | \Psi^- \rangle = 99.69(4)\%$, which is in good agreement with our analysis (see Supplementary Note 4), and demonstrates the preparation of high-fidelity entangled photon pairs. To the best of our knowledge, the photon–photon quantum logic gate demonstrated here has the highest truth table fidelity and entangling gate fidelity reported until now.

To further verify the entanglement, we demonstrate the violation of Bell's inequality with the output state by evaluating the correlation

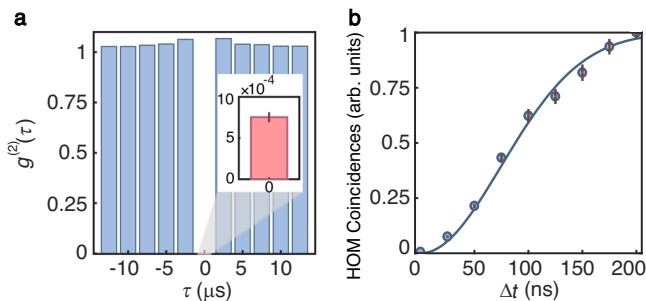

**Fig. 2 | High-quality Rydberg single-photon source. a** Second-order intensity correlation function $g^{(2)}(\tau)$ as a function of delay $\tau$. The duration of each experimental cycle is 2.5 μs. The insert shows $g^{(2)}(0)$ at zero delay. A 200-ns detection window is applied. **b** The normalized coincidences in HOM experiment as a function of two-photon temporal mismatch $\Delta t$. The solid curve is a Gaussian fit. The error bars represent the 1σ standard deviation from photoelectric counting events.

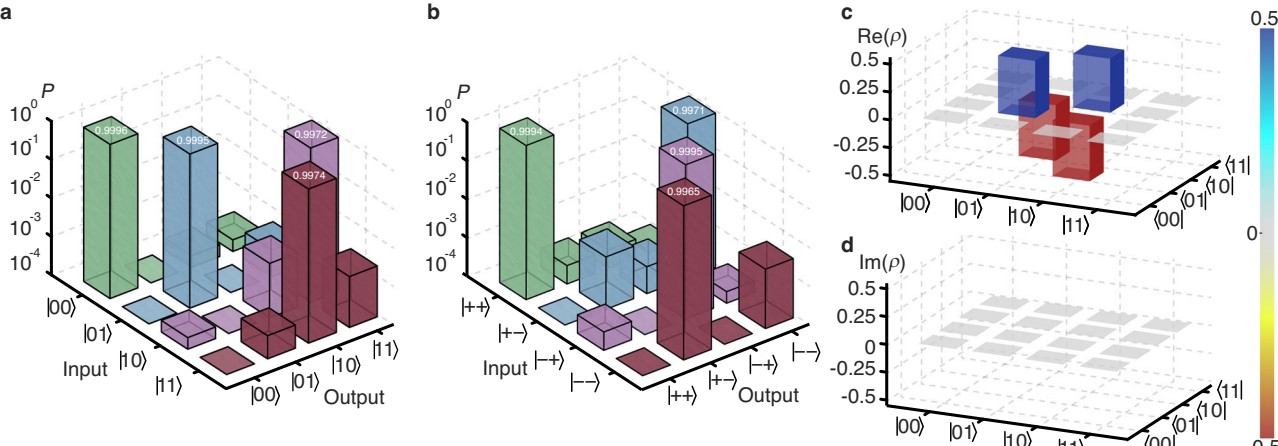

**Fig. 3 | High-fidelity photon–photon gate. a, b** Truth tables of the CNOT gate in the computational ZZ basis (**a**) and the complementary XX basis (**b**). The ideal input-output relation in the ZZ basis is $|00\rangle \to |00\rangle$, $|01\rangle \to |01\rangle$, $|10\rangle \to |11\rangle$ and $|11\rangle \to |10\rangle$. Similarly, the ideal input-output relation in the XX basis is $|++\rangle \to |++\rangle$, $|+-\rangle \to |--\rangle$, $|-+\rangle \to |-+\rangle$ and $|--\rangle \to |+-\rangle$, with the definitions of $|\pm\rangle_{\mathrm{C}} = (|0\rangle_{\mathrm{C}} \pm |1\rangle_{\mathrm{C}})/\sqrt{2}$ for control qubit and $|\pm\rangle_{\mathrm{T}} = (|0\rangle_{\mathrm{T}} \pm |1\rangle_{\mathrm{T}})/\sqrt{2}$ for target qubit. The measured probabilities are shown on a logarithmic scale. **c, d** Real (**c**) and imaginary (**d**) parts of the reconstructed density matrix $\rho$ for the created entangled state.

function $E(\theta_c, \theta_t)$ given by

$$\frac{C_{01}(\theta_c,\theta_t) + C_{10}(\theta_c,\theta_t) - C_{00}(\theta_c,\theta_t) - C_{11}(\theta_c,\theta_t)}{C_{01}(\theta_c,\theta_t) + C_{10}(\theta_c,\theta_t) + C_{00}(\theta_c,\theta_t) + C_{11}(\theta_c,\theta_t)}, \qquad (1)$$

where $\theta_c$ ($\theta_t$) is the polarization angle for the measurement of the control (target) output state, and $C_{ij}$ is the coincidence rate between SPCMs $c_{i=0,1}$ and $t_{j=0,1}$ in Fig. 1b. Correlation function $E(\theta_c, \theta_t)$ with high fringe visibilities is observed in Fig. 4. The values of $E(\theta_c, \theta_t)$ at polarization angle settings for Bell's inequality are shown in Table 1, from which the Bell parameter $S = E(\pi/8, 0) + E(\pi/8, \pi/4) + E(-\pi/8, 0) - E(-\pi/8, \pi/4)$ is determined. The measured value $S = 2.823(12)$ approaches the ideal value of $|S| = 2\sqrt{2}$, and violates the Clauser–Horne–Shimony–Holt inequality $|S| \leq 2$ by more than 60 standard deviations, thus clearly confirming the entanglement between the output photons and the quantum nature of our gate.

To quantitatively understand the performance of our quantum logic gate, we thoroughly investigate possible sources of error in the experiment, and identify three major contributions to the CNOT gate infidelity. Background detection events, which mainly come from the dark counts of SPCMs, occur randomly and lead to an infidelity of 0.088(4)% for ZZ basis. Moreover, the residual multi-photon components from the single-photon source induce unexpected coincidences in $|10\rangle$ and $|11\rangle$ bases, and lead to an infidelity of 0.067(19)% for ZZ basis. In addition, the imperfect photon indistinguishability reduces quantum interference visibility and causes errors during the controlled state flip of the target qubit, leading to an infidelity of 0.06(8)% for ZZ basis. The infidelity for XX basis contributed by the above-mentioned mechanisms is comparable. We emphasize that the above-mentioned error mechanisms are not intrinsic to our experiment, and can be mitigated in the future (see Supplementary Note 3).

The gate performance with 200 ns single-photon detection window is also studied, and the truth table fidelity for XX (ZZ) basis is 99.40(3)% (99.53(3)%). The dominating contribution of infidelity comes from the finite indistinguishability of 99.55(9)%, which is most likely caused by the phase chirps in the rising and falling edges of the single-photon temporal profile. In principle, this effect can be characterized by using a homodyne method to extract the single-photon phase profile and can be compensated by optimally controlling the phase and amplitude of the readout field $\Omega_{479}^r$.

## Table 1 | Violation of Bell's inequality

| $\theta_c$ | $\theta_t$ | $E(\theta_c, \theta_t)$ |
|---|---|---|
| $\pi/8$ | 0 | 0.706(6) |
| $\pi/8$ | $\pi/4$ | 0.714(6) |
| $-\pi/8$ | 0 | 0.702(6) |
| $-\pi/8$ | $\pi/4$ | −0.700(6) |
| | | $S = 2.823(12)$ |

The error bar represents the 1σ standard deviation.

The probabilistic linear-optical gate protocol employed here has an intrinsic efficiency of 1/9, and the probability of having a single photon at the target (control) input is about 7.1% (3.5%) with a 200-ns detection window. Considering the gate protocol efficiency, the single-photon efficiencies, optical losses, and the SPCM detection efficiency, the typical coincidence detection probability is about $7.6 \times 10^{-5}$. The 80-ns detection window further reduces it to $3.3 \times 10^{-5}$. With a repetition rate of 50 kHz, our experiment features a high success rate up to 100 per minute even with the 80-ns small detection window. This demonstrates quantum optical operation based on Rydberg single-photon source is competitive in gate success rate to other single-photon sources, while featuring further advantages in single-photon purity, indistinguishability, and quantum logic gate fidelity. The coincidence detection probability and repetition rate can be further improved with future technical efforts (see Supplementary Note 3).

We emphasize that the achieved high fidelity is not limited to the probabilistic gate scheme used here. For example, deterministic photon–photon gate protocols have been demonstrated using matter-light interactions[31,32], and a major source of infidelity in these experiments comes from the detrimental multi-photon components in the photonic qubits, which can be circumvented with our near-optimal single photons.

## Discussion

In summary, we experimentally demonstrate a photon–photon quantum logic gate with a post-selected truth table fidelity of 99.84(3)% and entangling gate fidelity of 99.69(4)% based on near-optimal single photons generated by Rydberg atoms. When combined with multiplexing techniques and quantum error correction, the demonstration of high-fidelity gate enables the reduction of quantum resource overhead and thus constitutes an important step toward building a large-scale linear-optical quantum computer.

Our results open up new perspectives for highly demanding applications such as photonic quantum information processing and distributed matter-light quantum architectures[33] (see Supplementary Note 5 for detailed discussions on potential applications). For example, high-fidelity single photons and two-photon gates enable the preparation of photonic cluster states, which are the key elements of the all-optical quantum repeaters[34,35] that circumvent the requirement of long-lived quantum memories. By increasing the size of the cluster states and extending their dimensions to higher than two, fault-tolerant photonic quantum computing can be implemented[6]. Furthermore, high-quality single photons from Rydberg atoms can be injected into integrated photonic chips[36], realizing photonic quantum circuits with excellent multi-photon quantum interference. Last, high-quality single photons and entangled photon pairs enable near-perfect interconnections between distant quantum modules[37,38] that employ atomic qubits as local quantum processors, building up large-scale quantum architectures[39] with low overall error rates.

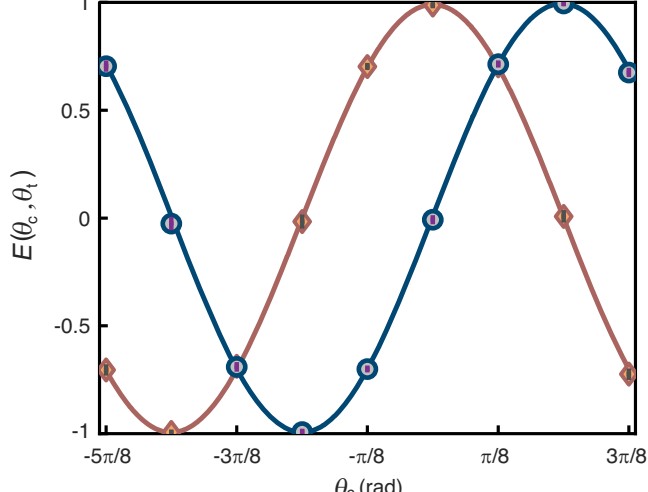

**Fig. 4 | Entanglement created by the quantum logic gate.** Measured correlation function $E(\theta_c, \theta_t)$ as a function of $\theta_c$ with $\theta_t = 0$ (diamonds) and $\theta_t = \pi/4$ (circles). Solid curves are sinusoidal fits with 0.99(1) visibility. The error bars represent the 1σ standard deviation from photoelectric counting events.

## Methods
### Rydberg excitation
To prepare the atomic sample, $^{87}$Rb atoms are loaded from the background vapor into a magneto-optical trap (MOT) for 330 ms. Then, the

atomic density is increased during the next 50 ms by compressing the MOT with a magnetic field gradient of 50 G cm$^{-1}$. Polarization gradient cooling further lowers the atomic temperature to about 10 μK and the atoms are loaded into a 1012-nm wavelength optical dipole trap. The dipole trap is formed by focusing a linearly polarized laser field, with a transverse waist of 10 μm and a vertical waist of 50 μm, respectively. In the next 50 ms the untrapped atoms leave the experimental region through free-fall, and a bias magnetic field of 8 G is applied to define the quantization axis perpendicular to the propagation direction of 1012 nm dipole trap. To initialize the atoms in the ground state $|g\rangle = |5S_{1/2}, F=2, m_F=2\rangle$, a $\sigma^+$-polarized optical pumping field is used to drive the $|5S_{1/2}, F=2\rangle \leftrightarrow |5P_{1/2}, F=2\rangle$ transition, and a repumper light resonant with $|5S_{1/2}, F=1\rangle \leftrightarrow |5P_{3/2}, F=2\rangle$ depletes the atoms from the hyperfine level $|5S_{1/2}, F=1\rangle$. The atoms are efficiently prepared in the ground state $|g\rangle$ within 230 μs and the atomic sample has an optical depth of ~5 for the laser field resonant with $|g\rangle \leftrightarrow |e\rangle$ transition.

## Single-photon generation

To generate single photons, the 2.5-μs-long experimental protocol is repeated 50,000 times after every atomic sample preparation. In each experimental cycle, two-photon Rydberg excitation is performed using a $\sigma^+$-polarized 780-nm laser field and a $\sigma^-$-polarized 479-nm laser field through the intermediate state $|e\rangle = |5P_{3/2}, F=3, m_F=3\rangle$ with a detuning of $\Delta/2\pi = -200$ MHz. The 780 and 479 nm excitation lasers off-resonantly couple $|g\rangle \leftrightarrow |e\rangle$ and $|e\rangle \leftrightarrow |r\rangle = |90S_{1/2} J=1/2, m_J=1/2\rangle$ transitions, respectively. Ideally, Rydberg states with higher principal quantum number $n$ feature stronger interactions and better multi-excitation suppression. However, detrimental effects such as long-lived Rydberg contaminants, ambient electric fields induced level shifts, and density-dependent dephasing also worsen with higher $n$. To balance the pros and cons, Rydberg state with $n=90$ is employed here for the generation of high-quality single photons. The 780-nm laser field is generated from an external cavity diode laser and the 479-nm laser field is produced by a second-harmonic generator that is seeded by power-amplified 959-nm laser light. The 780 and 959 nm lasers are frequency-locked to an ultra-low expansion cavity with a finesse of 20,000 and the linewidths of both lasers are below 10 kHz. The Rabi frequencies of the 780 and 479 nm excitation fields are $\Omega_{780}^e/2\pi \approx 6.4$ MHz and $\Omega_{479}^e/2\pi \approx 4.2$ MHz, respectively. A 350-ns-long collective $\pi$ pulse with 780 and 479 nm excitation lasers promotes the atoms from the ground state $|G\rangle$ into the collective single excitation state $|R\rangle$. After a storage period of 300 ns, a 479-nm readout laser field ($\Omega_{479}^r$) resonant with the $|r\rangle \leftrightarrow |e\rangle$ transition is turned on, and converts the state $|R\rangle$ into a single-photon field. As a result of collective emission, the generated single photons have the same spatial mode as the 780-nm excitation laser and can be conveniently coupled into a single-mode fiber. A gating acousto-optic modulator is used before the fiber to protect the SPCMs from the strong 780 nm excitation laser pulse.

## Data availability

Data supporting the plots within this paper are available through Zenodo at https://doi.org/10.5281/zenodo.6552691. Further information is available from the corresponding author upon reasonable request.

## Code availability

The code used in this study is available from the corresponding authors upon reasonable request.

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

## Acknowledgements

The authors thank Chao-Yang Lu, Timothy Ralph, Kang Tan, Yiqiu Ma, and Yijia Zhou for valuable discussions. This work was supported by the National Key Research and Development Program of China under Grants No. 2021YFA1402003, the National Natural Science Foundation of China (Grant No. U21A6006, No. 12004127, No. 12005067, and No. 12104173), and the Fundamental Research Funds for the Central Universities, HUST (Grant No. 5003012068).

## Author contributions

S.S. and L.L. conceived the idea. S.S., B.X., Y.L., and J.W. built the experimental setup. S.S., B.X., G.-S.Y., D.-S.X., and K.Z. performed the experiment and data analysis. K.Z. and D.S. accomplished theoretical calculation and error estimation. L.L. supervised the experiment. The manuscript was written through contributions from all authors.

## Competing interests

The authors declare no competing interests.
