## [Peer Review File · Nature Communications]

REVIEWERS' COMMENTS

Reviewer #1 (Remarks to the Author):

I thank the authors for taking the time to consider my questions. I am generally satisfied by their responses. I will only note that I still do not think the meaning of the x-axis of Fig. 2b is clear from the text--one really has to look at Fig. S2 to understand it. The authors should state clearly in the text that Δt is generated by adjusting the timing of the excitation pulses so the wavepackets do not arrive at the detector at the same time.

Reviewer #2 (Remarks to the Author):

I have now examined the revisions listed by the authors, and for the most part I am satisfied that the manuscript accurately conveys the science & underlying physics. As such I am comfortable supporting the publication in Nature Communications, once one final concern is addressed:

-The story in the abstract seems logically disjoint. They begin by saying "However, low fidelity two-photon quantum logic gates and their probabilistic nature result in a large resource overhead for fault tolerant quantum computation. Improving the photonic quantum gate fidelity and efficiency alleviates the overwhelming resource consumption and holds the promise to tackle the probabilistic problem in linear optical entangling operations with error correction." Then move on to what appears to be a connected statement "A major limitation on the fidelity of photonic quantum logic gates is the imperfection of single photons. Here, we overcome this obstacle by using high-quality single photons generated from Rydberg atoms as qubits for the interference-based linear optical gate protocol, and achieve..." This would all be fine, except that what they then quote are post selected gate fidelities, which only depend upon the single photons being high-quality insofar as their g_2 s are small (that is- we aren't background dominated). Indeed, a less-than-educated reader might not notice the "post-selected" in the >99% efficiency. I think thus think it is important to quote the actual unconditional success probability (not the rate in events/min), and state how big an improvement it is over prior

works; if what is actually improved is the successful gate _rate_, then the abstract should be written in a way that motivates that being important, reveals that number, and compares it to the best prior work.

Reviewer #3 (Remarks to the Author):

The authors have strongly modified their manuscript and incorporated the main criticism in the first round. In addition, the manuscript is transferred to nature communication, which in the referee's opinion is the appropriate journal to present their results. The analysis is performed carefully and presents a technical and challenging improvement over previous approaches. Therefore, the manuscript can be published as it is.

Response to Referee #1

I thank the authors for taking the time to consider my questions. I am generally satisfied by their responses. I will only note that I still do not think the meaning of the x-axis of Fig. 2b is clear from the text—one really has to look at Fig. S2 to understand it. The authors should state clearly in the text that Δt is generated by adjusting the timing of the excitation pulses so the wavepackets do not arrive at the detector at the same time.

Response: We thank the referee for reviewing our manuscript and for the positive comments. To clarify the meaning of x-axis in Fig. 2b, we have added a few sentences in the manuscript (starting from line 137):

‘The two photons used in the HOM experiment are sequentially generated from the Rydberg atoms. The first photon is delayed by $5\ \mu\text{s}$ using an EOM and a 1 km fiber, while the second photon is generated $5\ \mu\text{s} + \Delta t$ after the first one (see Supplementary Note 2).’

Response to Referee #2

I have now examined the revisions listed by the authors, and for the most part I am satisfied that the manuscript accurately conveys the science & underlying physics. As such I am comfortable supporting the publication in Nature Communications, once one final concern is addressed:

-The story in the abstract seems logically disjoint. They begin by saying “However, low fidelity two-photon quantum logic gates and their probabilistic nature result in a large resource overhead for fault tolerant quantum computation. Improving the photonic quantum gate fidelity and efficiency alleviates the overwhelming resource consumption and holds the promise to tackle the probabilistic problem in linear optical entangling operations with error correction.” Then move on to what appears to be a connected statement “A major limitation on the fidelity of photonic quantum logic gates is the imperfections of single photons. Here, we overcome this obstacle by using high-quality single photons generated from Rydberg atoms as qubits for the interference-based linear optical gate protocol, and achieve...” This would all be fine, except that what they then quote are post-selected gate fidelities, which only depend upon the single photons being high-quality in so far as their $g^{(2)}$ s are small (that is -we aren’t background dominated). Indeed a less-than-educated reader might not notice the “post-selected” in the $> 99\%$ efficiency. I think thus think it is important to quote the actual unconditional success probability (not the rate in events/min), and state how big an improvement it is over prior works; if what is actually improved is the successful gate rate, then the abstract should be written in a way that motivates that being important, reveals that number, and compares it to the best prior work.

Response: We thank the referee for reviewing our manuscript and for the positive comments. We have modified the abstract based on referee’s suggestion.

In the second version of the manuscript, we wanted to point out in the abstract that, although the high success probability and fidelity of quantum logic gate are both essential for all-optical quantum computation, our work is not aimed at improving the low gate success probability, which can, in principle, be solved by some of the existing methods, such as multiplexing. Instead, we focus on tackling the low gate fidelity issue, which is previously limited by the imperfections of single-photon source and can now be greatly improved with our near-optimal Rydberg single-photon source.

However, as the referee suggested, the way current abstract was written might still confuse the readers about the goal of our work. Therefore, we have revised the abstract to emphasize that our work focuses mainly on improving the quantum logic gate fidelity instead of gate success probability. The revised abstract is shown in the following (changes have been highlighted):

‘Compared to other types of qubits, photon is one of a kind due to its unparalleled advan-

tages in long-distance quantum information exchange. Therefore, photon is a natural candidate for building a large-scale, modular optical quantum computer operating at room temperature. However, low-fidelity two-photon quantum logic gates and their probabilistic nature result in a large resource overhead for fault tolerant quantum computation. While the probabilistic problem can, in principle, be solved by employing multiplexing and error correction, the fidelity of linear-optical quantum logic gate is limited by the imperfections of single photons. Here, we report the demonstration of a linear-optical quantum logic gate with truth table fidelity of 99.84(3)% and entangling gate fidelity of 99.69(4)% post-selected upon the detection of photons. The achieved high gate fidelities are made possible by our near-optimal Rydberg single-photon source. Our work paves the way for scalable photonic quantum applications based on near-optimal single-photon qubits and photon-photon gates.'

We agree with the referee that non-specialists might not notice or understand the “post-selected” in the $> 99\%$ fidelity. Therefore, we have changed the sentence to: ‘Here, we report the demonstration of a linear-optical quantum logic gate with truth table fidelity of 99.84(3)% and entangling gate fidelity of 99.69(4)% post-selected upon the detection of photons.’ Now the readers should be able to clearly understand that the measured results are ‘post-selected upon the detection of photons’.

As for the comparison of our work to previous ones, we have described prior state-of-the-art linear-optical gate experiments in the introduction and provided quantitative comparison on fidelities in Supplementary Note 3. We agree that gate success probability is important for practical applications and we have analyzed our unconditional gate success probability in the manuscript (starting from line 247) and Supplementary Note 3. However, pushing the limit of gate success probability is not the aim of this work. Therefore, we feel it will be easier for the readers to understand the goal of the paper (improving the gate fidelity), if the abstract focuses on fidelity instead of gate success probability.

Response to Referee #3

The authors have strongly modified their manuscript and incorporated the main criticism in the first round. In addition, the manuscript is transferred to nature communication, which in the referee’s opinion is the appropriate journal to present their results. The analysis is performed carefully and presents a technical and challenging improvement over previous approaches. Therefore, the manuscript can be published as it is.

Response: We thank the referee for reviewing our manuscript and for the positive comments.